# Fatty acid binding protein 5 regulates docetaxel sensitivity in taxane-resistant prostate cancer cells

Andrew Hillowe[1], Chris Gordon[1], Liqun Wang[1], Robert C. Rizzo[2], Lloyd C. Trotman[3], Iwao Ojima[4,5], Agnieszka Bialkowska[5,6], Martin Kaczocha[1,5]*

**1** Department of Anesthesiology, Renaissance School of Medicine, Stony Brook University, Stony Brook, New York, United States of America, **2** Department of Applied Mathematics and Statistics, Stony Brook University, Stony Brook, New York, United States of America, **3** Cold Spring Harbor Laboratory, Cold Spring Harbor, New York, United States of America, **4** Department of Chemistry, Stony Brook University, Stony Brook, New York, United States of America, **5** Institute of Chemical Biology and Drug Discovery, Stony Brook University, Stony Brook, New York, United States of America, **6** Department of Medicine, Renaissance School of Medicine, Stony Brook University, Stony Brook, New York, United States of America

* Martin.Kaczocha@Stonybrook.edu

**Data Availability Statement:** All data are available from the Figshare database (DOI: https://doi.org/10.6084/m9.figshare.22304824.v1).

## Abstract

Prostate cancer is a leading cause of cancer-related deaths in men in the United States. Although treatable when detected early, prostate cancer commonly transitions to an aggressive castration-resistant metastatic state. While taxane chemotherapeutics such as docetaxel are mainstay treatment options for prostate cancer, taxane resistance often develops. Fatty acid binding protein 5 (FABP5) is an intracellular lipid chaperone that is upregulated in advanced prostate cancer and is implicated as a key driver of its progression. The recent demonstration that FABP5 inhibitors produce synergistic inhibition of tumor growth when combined with taxane chemotherapeutics highlights the possibility that FABP5 may regulate other features of taxane function, including resistance. Employing taxane-resistant DU145-TXR cells and a combination of cytotoxicity, apoptosis, and cell cycle assays, our findings demonstrate that FABP5 knockdown sensitizes the cells to docetaxel. In contrast, docetaxel potency was unaffected by FABP5 knockdown in taxane-sensitive DU145 cells. Taxane-resistance in DU145-TXR cells stems from upregulation of the P-glycoprotein ATP binding cassette subfamily B member 1 (ABCB1). Expression analyses and functional assays confirmed that FABP5 knockdown in DU145-TXR cells markedly reduced ABCB1 expression and activity, respectively. Our study demonstrates a potential new function for FABP5 in regulating taxane sensitivity and the expression of a major P-glycoprotein efflux pump in prostate cancer cells.

## Introduction

Prostate cancer (PC) remains the second leading cause of cancer-related death in men in the United States [1]. Mainstay treatments include anti-androgen and microtubule-stabilizing chemotherapeutics such as docetaxel designed to limit androgenic signaling and thwart tumor

**Funding:** This study was funded by the National Cancer Institute grant #CA237154. The funders had no role in study design, data collection and analysis, decision to publish, or preparation of the manuscript.

**Competing interests:** The authors have declared that no competing interests exist.

progression [2]. However, tumors invariably transition into castration-resistant prostate cancer (CRPC) that is unresponsive to androgen deprivation therapies and can likewise acquire taxane resistance. Taxane resistance stems from distinct cellular processes including mutations in the tubulin protein that compromise taxane binding, alterations in intracellular signaling pathways, and upregulation of drug efflux transporters including the P-glycoprotein ATP binding cassette subfamily B member 1 (ABCB1), also known as multidrug resistance protein 1 (MDR1) [3].

Fatty acid binding proteins (FABPs) are a family of ten structurally related cytoplasmic proteins that shuttle fatty acids and related bioactive lipids to intracellular organelles and nuclear receptors [4]. While FABPs are not expressed in the healthy human prostate, FABP5 becomes highly upregulated in advanced PC [5–8]. *FABP5* is one of fifteen signature genes whose upregulation can predict PC and is prominently featured in metastatic PC [9–12]. This expression pattern is likewise observed in PC cells, with weakly aggressive cell lines lacking FABP5 expression while highly aggressive cells demonstrate high FABP5 expression levels [13,14]. Mechanistically, FABP5 enhances PC progression through several mechanisms including enhancement of lipid metabolism, increased survival, and trafficking of lipid ligands to nuclear peroxisome proliferator-activated receptor gamma, resulting in receptor activation and upregulation of proangiogenic and pro-metastatic genes [13,15–20].

Pharmacological or genetic inhibition of FABP5 dampens peroxisome proliferator-activated receptor gamma activation, reduces tumor growth, and attenuates the metastatic potential of prostate tumors [15,16,21–23]. These results, combined with clinical evidence demonstrating elevated FABP5 expression in metastatic PC, highlight the potential for FABP5 inhibitors as anticancer therapeutics [8]. Prior work from our group showed that small molecule inhibitors of FABP5 synergize with taxane therapeutics to reduce prostate tumor growth [21]. Since taxane resistance often accompanies advanced PC, it is important to determine whether FABP5 expression influences taxane resistance. In the present study, we employ taxane-resistant DU145 cells (DU145-TXR) characterized by ABCB1 overexpression [24] and demonstrate that FABP5 knockdown reduces ABCB1 expression and function, and resensitizes the cells to the chemotherapeutic docetaxel.

## Materials and methods

### Cell lines

DU145-TXR cells were kindly provided by Dr. Ram Mohato from the University of Nebraska Medical Center and were previously described [24]. Parental DU145 cells were purchased from ATCC and both DU145-TXR and DU145 lines were maintained in RPMI1640 (Thermo Fisher) supplemented with 10% fetal bovine serum (FBS) (VWR Life Science) and 100 units/mL of penicillin/streptomycin (Thermo Fisher) in a humidified incubator at 37˚C with 95% $O_2$ and 5% $CO_2$. DU145-TXR cells were maintained in media containing 200 nM paclitaxel (Sigma-Aldrich).

### MTT cytotoxicity assay

Cell viability was assessed using the 3-(4,5-dimethylthiazol-2-yl)- 2,5-diphenyltetrazolium bromide (MTT) colorimetric assay (Sigma-Aldrich) as we previously described [21]. DU145 and DU145-TXR cells were seeded into 96-well plates (Sarstedt) at $5 \times 10^3$ cells/well and incubated with 0.0003 nM to 300 nM docetaxel (Sigma-Aldrich) or vehicle (0.1% DMSO) in RPMI 1640 containing 1% FBS for 24 to 72 h. Media were removed, and cells washed with PBS (Gibco) and subsequently treated with 0.5 mg/ml MTT in serum-free RPMI 1640 for 4 h. The cells were then solubilized in 100% DMSO for 1 h and the absorbance of each well was measured at

a wavelength of 562 nm in an F5 Filtermax Multi-Mode Microplate Reader (Molecular Devices).

## FABP5 knockdown

DU145 and DU145-TXR cells were transfected with FABP5 shRNA clone V3LHS_402771 or scrambled control and stable cells established using 1 μg/ml puromycin (Gibco) as we previously described [15,25]. Populations of cells expressing GFP were identified using flow cytometry, sorted, and subsequently propagated in media containing puromycin.

## AnnexinV and propidium iodide labeling

AnnexinV and propidium iodide (PI) kits were from Thermo-Fisher. Cells ($5x10^4$) were plated onto 6-well tissue culture plates (Sarstedt) 24 h prior to docetaxel dosing in complete RPMI-1640 media containing 5% FBS. Cells were incubated with vehicle (0.1% DMSO), 3 or 30 nM docetaxel in complete RPMI-1640 for 24, 48 or 72 h. The cells were then collected and resuspended in ice-cold PBS and maintained on ice for 30 min. The cells were pelleted and processed according to the manufacturer's protocol (Thermo Fisher). AnnexinV and PI signals were acquired on a CytoFlex LX instrument (Beckman Coulter).

## Western blotting

Western blotting was performed as previously described [26,27].

The following antibodies were used: mouse anti-β-actin (Cell Signaling, #3700), rabbit anti-FABP5 (BioVendor, #RD181060100), and anti-ABCB1 (Cell Signaling, #12683).

## qPCR

RNA extractions and qPCR were performed as described [28]. The following primers were used: *ACTB*/β-actin (Forward: `GACGGCCAGGTCATCACTAT`, Reverse: `CGGATGTCAACG TCACACTT`), FABP5 (Forward: `TCAGCAGCTGGAAGGAAGAT`, Reverse: `TTCGCAAAGCTA TTCCCACT`), and ABCB1 (Forward: `GCTGTCAAGGAAGCCAATGCCT`, Reverse: `TGCAATGG CGATCCTCTGCTTC`). Quantification was performed using the $2^{-\Delta\Delta Ct}$ method with β-actin serving as the housekeeping gene.

## Cell cycle analysis

Cells ($5x10^4$) were treated with vehicle or docetaxel for 72 h and were then washed with ice-cold PBS, trypsinized, pelleted, resuspended in fixation buffer (70% ethanol in PBS), and stored at -20˚C for 24–72 h. Next, 5 ml of staining buffer (1% BSA, 0.2% Triton-X in PBS) was added to each sample. Following centrifugation, the staining buffer was removed and each sample resuspended in 200 μl of FACS buffer (1mg/ml RNase, 0.1% Triton-X100 in $H_2O$) and transferred to polystyrene round-bottom tubes containing cell-strainer caps (Falcon #352235). PI solution (2 mM in $H_2O$) was then added directly to each tube and quantification was performed on a CytoFLEX LX instrument (Beckman Coulter).

## MDR1 activity assay

MDR1 activity was quantified using the Multidrug Resistance Direct Dye Efflux Assay (Sigma, #ECM910). Briefly, cells from each treatment condition were trypsinized, collected by centrifugation, resuspended in 3,3'-Diethyloxacarbocyanine iodide (DiOC2(3)), and incubated on ice for 15 min. The cells were then collected by centrifugation and resuspended twice in 2.5 ml ice-cold efflux buffer (RPMI-1640 containing 1% FBS) per $10^6$ cells. The cells were then

incubated in warm efflux buffer containing 22 μM vinblastine or 0.1% DMSO (37˚C), or at 4˚C in efflux buffer for 30 min. The reactions were then quenched twice with 5 ml of ice-cold efflux buffer followed by centrifugation at 4˚C. Finally, the cells were resuspended in 300 μl of cold efflux buffer and 100 μl was transferred to a 96-well plate. Fluorescence was measured on an F5 Filtermax Multi-Mode Microplate Reader (Molecular Devices) with excitation and emission wavelengths of 530 nm and 585 nm, respectively. Percent inhibition of MDR1 activity was quantified as follows: ([Vinblastine-Vehicle]/[4˚C-Vehicle]*100).

### Statistical analysis

Western blots and qPCR results were analyzed using unpaired t-test or One-Way ANOVA followed by Dunnett's post hoc test as appropriate. Cell viability, apoptosis, and cell cycle results were analyzed by One-Way ANOVA followed by Dunnett's post-hoc test.

## Results

### *FABP5* knockdown sensitizes DU145-TXR cells to docetaxel

We first confirmed that DU145 and DU145-TXR cells express FABP5 and established corresponding FABP5 knockdown (KD) lines (Fig 1A–1C). MTT assays revealed that docetaxel induced cytotoxicity in DU145 cells with an $IC_{50}$ of 6.1 nM (Fig 1D). In contrast, DU145-TXR cells were resistant to docetaxel and displayed cytotoxicity only at 300 nM (Fig 1D), the highest concentration tested due to the known solubility challenges with this compound. While FABP5 knockdown did not alter docetaxel sensitivity in DU145 cells ($IC_{50}$ of 5.3 nM), DU145-TXR KD cells became resensitized to docetaxel ($IC_{50}$ of 4.4 nM) (Fig 1D). These results indicate that FABP5 regulates docetaxel sensitivity in DU145-TXR cells.

### Induction of apoptosis by docetaxel in DU145 and DU145-TXR cells

To gain insights into the mode and temporal profile of cell death induction by docetaxel, we assessed apoptosis and cellular viability using Annexin V and propidium iodide (PI) labeling across the 72-h time course of docetaxel treatment. As expected, docetaxel reduced the viability and increased the proportion of apoptotic DU145 cells at the 24h, 48h, and 72h time points (Fig 2A, Table 1). A similar profile was observed in DU145 KD cells (Fig 2B, Table 1). In contrast, docetaxel failed to reduce the viability and induce apoptosis in DU145-TXR cells at comparable concentrations (Fig 3A, Table 1). Consistent with the MTT results, the knockdown of FABP5 resensitized DU145-TXR cells to docetaxel (Fig 3B, Table 1). The time course of apoptosis induction was similar between the cell-lines, with apoptosis evident as early as the 24h time point.

### Cell cycle analysis

We subsequently analyzed the cell cycle profiles of the four cell-lines after treatment with docetaxel. G0/G1, S and G2/M cell cycle phases were evident in vehicle-treated DU145 and DU145 KD cells while incubation with docetaxel induced a near complete loss of S phase characteristics, reduction in G0/G1, and a shift into G2/M (Fig 4A and 4B). Notably, DU145-TXR cells were resistant to the docetaxel-induced shift in the cell cycle as a decrease in the S phase and an increase in G2/M was not observed (Fig 4C). In contrast, DU145-TXR KD cells displayed a cell cycle profile comparable to the parental DU145 and DU145 KD cell-lines (Fig 4).

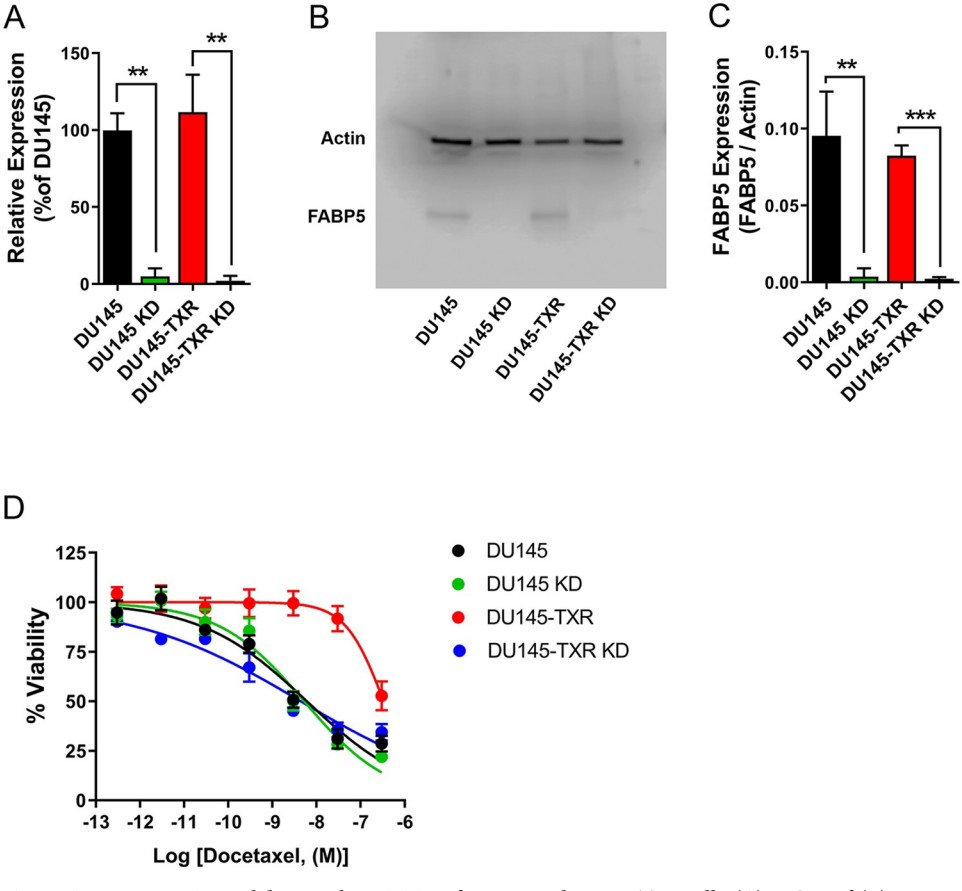

**Fig 1. FABP5 expression and docetaxel sensitivity of DU145 and DU145 TXR cells.** (**A**) qPCR and (**B**) western blotting demonstrates FABP5 expression in DU145 and DU145-TXR cells and confirms successful FABP5 KD in the corresponding cell-lines (n = 3). (**C**) Quantification of western blots presented as FABP5 / β-Actin (n = 3). (**D**) Cytotoxicity of docetaxel in taxane-resistant DU145-TXR and taxane-sensitive DU145 cells and in cells bearing an FABP5 KD. The $IC_{50}$ values for docetaxel cytotoxicity were 6.1 nM, 5.3 nM, >100 nM, and 4.4 nM in DU145, DU145 KD, DU145-TXR, and DU145-TXR KD cells, respectively. **, $p < 0.01$; ***, $p < 0.001$ (n = 4). Data are reported as % viability of the respective vehicle controls.

## ABCB1 expression and activity in DU145-TXR and KD cells

Taxane resistance in DU145-TXR cells stems from overexpression of ABCB1 [24]. We examined the expression of ABCB1 in the four cell-lines and observed robust ABCB1 expression in DU145-TXR cells while the levels of ABCB1 were low in DU145 cells (Fig 5A and 5B). Compared to DU145-TXR cells, ABCB1 expression was reduced by ~98% in DU145-TXR KD cells (Fig 5A–5C), confirming downregulation of ABCB1 upon FABP5 knockdown. Next, we examined MDR1 activity via an assay that quantifies the efflux of the fluorescent ligand DiOC2(3) in the presence or absence of the selective MDR1 inhibitor vinblastine. Our results indicate that while vinblastine increased the retention of DiOC2(3) in DU145-TXR cells, consistent with MDR1 inhibition, a comparable effect was not observed in DU145-TXR KD cells (Fig 5D), confirming nonfunctional MDR1. Taken together, our results demonstrate that the knockdown of FABP5 reduces ABCB1 expression in taxane-resistant DU145 cells and sensitizes the cells to docetaxel.

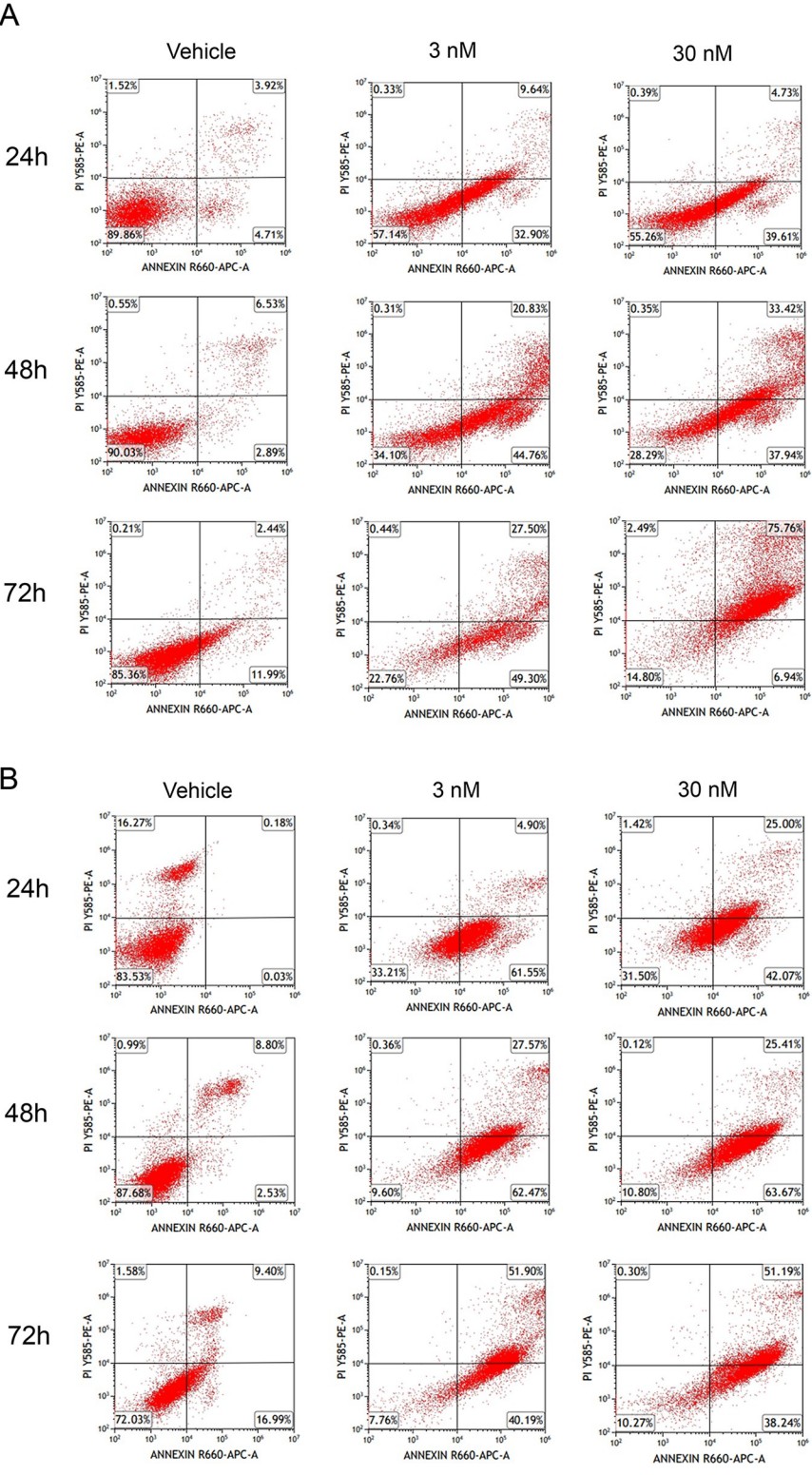

**Fig 2. Effects of docetaxel on apoptosis and cellular viability in DU145 and DU145 KD cells.** (**A**) DU145 and (**B**) DU145 KD cells were incubated with vehicle (0.1% DMSO), 3 nM, or 30 nM docetaxel for 24, 48 or 72 h. Annexin V and PI staining was quantified at each time point. The percent of viable cells appears in the lower left quadrant while apoptotic cells appear in the upper and lower right quadrants.

**Table 1. Percent viable and apoptotic cells for DU145, DU145-TXR after DTX time course treatment.**

| Time | Docetaxel (nM) | DU145 | | DU145 KD | | DU145-TXR | | DU145-TXR KD | |
|---|---|---|---|---|---|---|---|---|---|
| | | **Viable** | **Apoptotic** | **Viable** | **Apoptotic** | **Viable** | **Apoptotic** | **Viable** | **Apoptotic** |
| **24h** | 0 | 90.0 ± 0.1 | 8.9 ± 0.4 | 80.7 ± 7.0 | 12.3 ± 12.9 | 90.3 ± 7.6 | 8.4 ± 8.1 | 85.5 ± 7.8 | 12.2 ± 9.0 |
| | 3 | 57.6 ± 1.3*** | 42.1 ± 1.4*** | 34.6 ± 2.4*** | 64.5 ± 2.9*** | 85.7 ± 4.1 | 13.4 ± 4.1 | 83.6 ± 9.2 | 14.8 ± 10.1 |
| | 30 | 55.3 ± 0.2*** | 44.2 ± 0.3*** | 31.3 ± 1.4*** | 67.6 ± 1.5*** | 80.3 ± 8.6 | 19.1 ± 8.8 | 60.4 ± 5.9* | 37.9 ± 4.5* |
| **48h** | 0 | 91.4 ± 2.8 | 6.1 ± 5.0 | 83.4 ± 10.4 | 15.5 ± 9.7 | 90.5 ± 7.1 | 8.2 ± 7.6 | 88.4 ± 1.4 | 9.6 ± 1.9 |
| | 3 | 33.1 ± 0.9*** | 66.7 ± 1.0*** | 10.4 ± 1.3*** | 89.3 ± 1.3*** | 78.7 ± 8.1 | 20.2 ± 8.3 | 66.4 ± 5.3** | 31.7 ± 4.3** |
| | 30 | 27.7 ± 2.0*** | 71.9 ± 2.0*** | 10.7 ± 1.5*** | 89.0 ± 1.6*** | 79.5 ± 9.5 | 19.6 ± 9.8 | 54.0 ± 6.8*** | 44.5 ± 6.7*** |
| **72h** | 0 | 85.4 ± 0.9 | 14.5 ± 0.9 | 76.4 ± 8.4 | 21.4 ± 9.2 | 84.4 ± 2.4 | 10.0 ± 6.9 | 84.4 ± 3.1 | 9.2 ± 8.5 |
| | 3 | 20.3 ± 2.3*** | 79.2 ± 2.2*** | 8.8 ± 1.4*** | 91.7 ± 1.4*** | 84.1 ± 3.0 | 7.1 ± 4.5 | 29.2 ± 11.8*** | 69.9 ± 11.7*** |
| | 30 | 14.4 ± 0.8*** | 83.2 ± 0.8*** | 10.7 ± 1.0*** | 89.0 ± 1.0*** | 84.1 ± 5.8 | 14.1 ± 6.4 | 39.5 ± 7.2*** | 59.3 ± 8.2** |

*, $p < 0.05$

**, $p < 0.01$

***, $p < 0.001$ vs vehicle (0) controls (n = 3).

## Discussion

Reprogramming of fatty acid signaling and metabolic pathways accompanies the progression of PC [29,30]. FABP5 is the principal FABP that is upregulated in advanced PC and is implicated as a critical driver of PC metastasis [5–8]. Pharmacological or genetic FABP5 inhibition reduces PC survival, tumor growth, and metastasis, thus positioning FABP5 as a promising therapeutic target [8,15,16,22,23]. Antiandrogen and taxane chemotherapeutics represent first line treatment options for PC [2]. We previously demonstrated that docetaxel, when combined with FABP5 inhibitors, produces enhanced suppression of tumor growth compared to either agent alone [21]. As prostate tumors often develop taxane resistance, in this study we sought to ascertain the impact of FABP5 upon docetaxel resistance in a taxane-resistant PC cell-line harboring ABCB1 overexpression.

Our results revealed that knockdown of FABP5 in DU145-TXR cells restored docetaxel sensitivity to levels observed in the parental taxane-sensitive DU145 cells. These resensitizing effects were evident across assays of cytotoxicity, apoptosis, and the cell cycle. The lack of sensitizing effects in DU145 KD cells confirmed that these effects stem from the modulation of processes or proteins unique to DU145-TXR cells. As ABCB1 overexpression underlies taxane resistance in DU145-TXR cells [24], we further demonstrated that FABP5 knockdown markedly reduced MDR1 expression and function in DU145-TXR KD cells.

Multiple mechanisms promote ABCB1 upregulation in PC cells including androgen receptors and retinoic acid receptor-related orphan receptor γ, which may operate independently or in concert [31,32]. Previous work in innate immune cells indicates that FABP5 regulates cytokine levels and downstream retinoic acid receptor-related orphan receptor γ expression [33], suggesting that a similar mechanism may operate in PC. NF-κB has additionally been implicated in promoting ABCB1 upregulation [34,35]. As FABP5 is known to regulate NF-κB function, inhibition of NF-κB following FABP5 knockdown could constitute a potential mechanism leading to reduced ABCB1 expression [36]. Several additional mechanisms have been proposed to upregulate ABCB1 including KRAS [37]. The vast number and diversity of pathways that contribute to ABCB1 expression highlights the possibility that FABP5 may maintain ABCB1 expression in taxane-resistant PC cells by engaging one or several key signaling pathways. Future studies will be required to elucidate the precise mechanism(s) underlying the regulation of ABCB1 expression by FABP5.

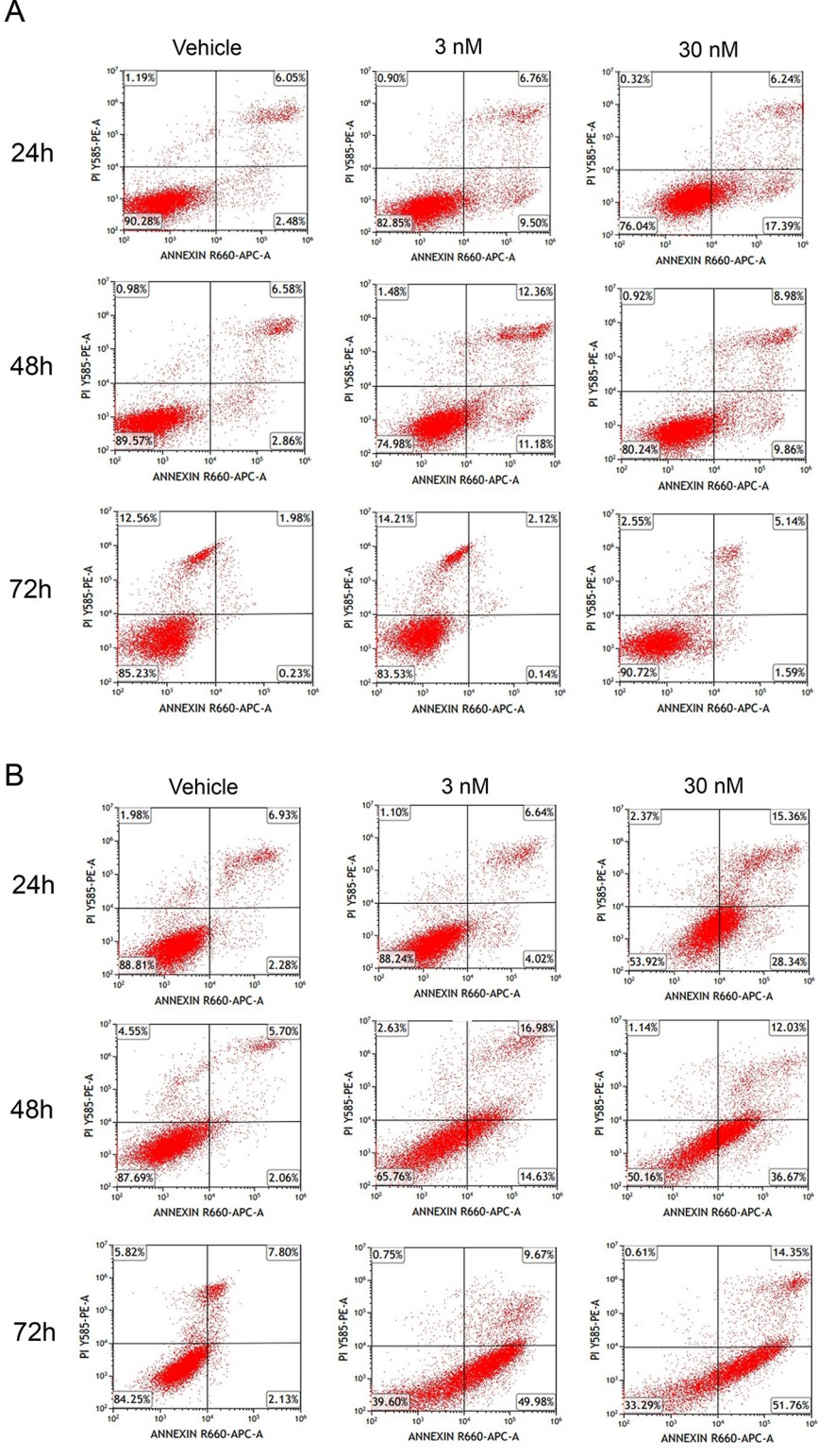

**Fig 3. Effects of docetaxel upon viability and apoptosis of DU145-TXR and DU145-TXR KD cells.** (**A**) DU145-TXR and (**B**) DU145-TXR KD cells were incubated with vehicle (0.1% DMSO), 3 nM, or 30 nM docetaxel for 24, 48 or 72 h, and Annexin V and PI staining was performed. The percent of viable cells appears in the lower left quadrant while apoptotic cells appear in the upper and lower right quadrants.

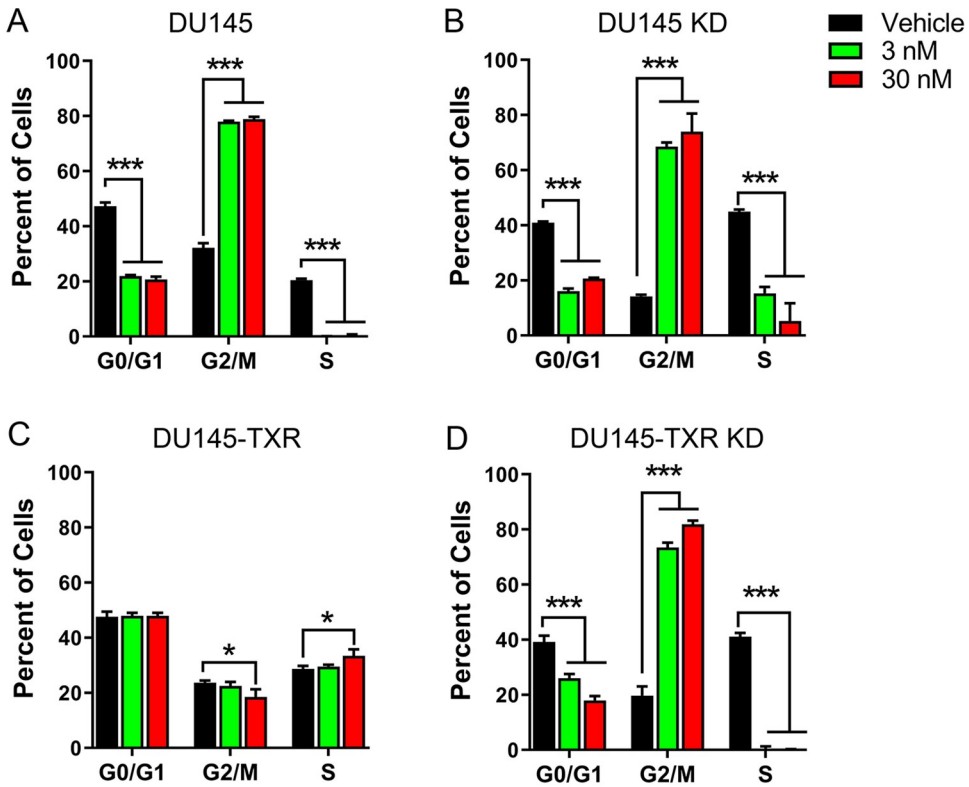

**Fig 4. Cell cycle profile in response to docetaxel treatment.** (**A**) DU145, (**B**) DU145 KD, (**C**) DU145-TXR, and (**D**) DU145-TXR KD cells were treated with vehicle (0.1% DMSO), 3 nM, or 30 nM docetaxel for 72 h and subsequently stained with PI and analyzed by flow cytometry. *, $p < 0.05$; ***, $p < 0.001$ (n = 3).

One limitation of the present study lies using a single taxane-resistant PC cell-line. Therefore, it will be important to assess if FABP5 modulation of ABCB1 expression and taxane resistance extends to other PC cell lines. Furthermore, as our study was performed entirely *in vitro*, it remains to be determined whether taxane resensitization will be observed *in vivo*. Notably, as our studies employed mixed populations of FABP5 knockdown cells, this mitigates the possibility that the effects observed herein originate from a single clone.

## Conclusions

This study demonstrates that FABP5 exerts a key role in maintaining the expression of ABCB1 in a PC cell-line, thereby ascribing a novel function to FABP5 in taxane resistance. Considering the known contribution of FABP5 to PC progression, these findings provide additional evidence that FABP5 inhibition is a sound therapeutic strategy to reduce tumor growth and metastasis and potentially blunt the development of taxane resistance.

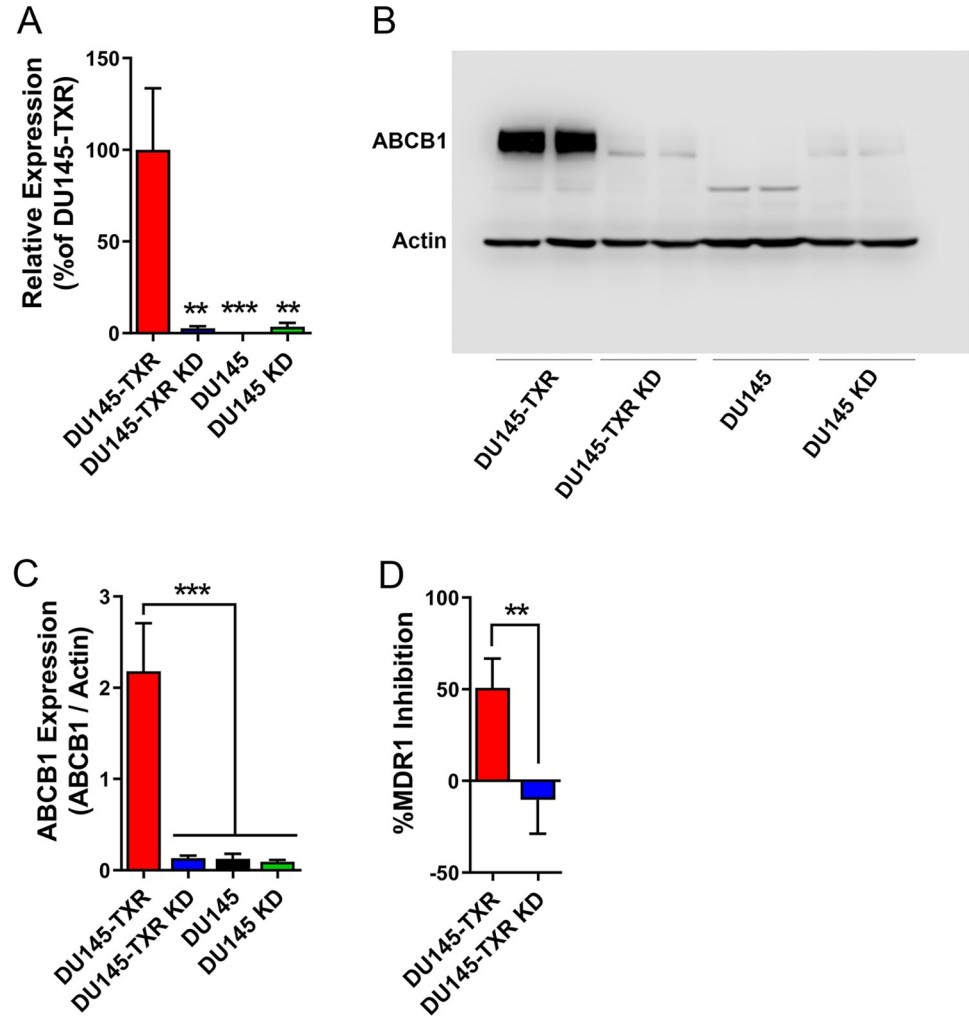

**Fig 5. Effect of FABP5 knockdown on ABCB1/MDR1 activity and function.** (**A**) qPCR analysis of ABCB1 expression in the four cell-lines (n = 3). (**B**) Western blot of ABCB1 expression. (**C**) Quantification of western blot results (n = 3). (**D**) MDR1 activity was quantified in DU145-TXR and DU145-TXR KD cells incubated with vehicle or the selective MDR1 inhibitor vinblastine (22 μM). Data are presented as DiOC2(3) efflux in cells incubated with vinblastine / vehicle controls. **, p < 0.01; ***, p < 0.001 (n = 4).

## Supporting information

**S1 Fig. Cell cycle analyses of DU145, DU145 KD, DU145-TXR, and DU145-TXR KD cells treated for 72h with vehicle, 3nM, or 30 nM docetaxel.** G0/G1, G2/M, and S phase populations were resolved on a linear scale using ModFit LX software. G0/G1 and G2/M correspond to a PI signal of 30 and 60%, respectively. S-phase is represented by the shaded segments while apoptotic cells are contained within the solid teal curves and assigned a PI signal of 10%. (TIF)

**S1 Raw images.**
(PDF)

## Acknowledgments

We would like to thank the Research Flow Cytometry Core in the Department of Pathology, Stony Brook University for assistance with data analysis.

## Author Contributions

**Conceptualization:** Andrew Hillowe, Robert C. Rizzo, Lloyd C. Trotman, Iwao Ojima, Agnieszka Bialkowska, Martin Kaczocha.

**Data curation:** Andrew Hillowe.

**Formal analysis:** Andrew Hillowe, Martin Kaczocha.

**Funding acquisition:** Robert C. Rizzo, Lloyd C. Trotman, Iwao Ojima, Martin Kaczocha.

**Investigation:** Andrew Hillowe, Chris Gordon, Liqun Wang, Agnieszka Bialkowska.

**Methodology:** Andrew Hillowe, Chris Gordon, Robert C. Rizzo, Lloyd C. Trotman, Agnieszka Bialkowska.

**Project administration:** Iwao Ojima.

**Supervision:** Agnieszka Bialkowska, Martin Kaczocha.

**Visualization:** Martin Kaczocha.

**Writing – original draft:** Andrew Hillowe, Lloyd C. Trotman, Iwao Ojima, Agnieszka Bialkowska, Martin Kaczocha.

**Writing – review & editing:** Andrew Hillowe, Robert C. Rizzo, Lloyd C. Trotman, Iwao Ojima, Agnieszka Bialkowska, Martin Kaczocha.

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
