## [Decision Letter · Decision Letter 0]

21 Sep 2023

Fatty Acid Binding Protein 5 Regulates Docetaxel Sensitivity in Taxane-Resistant Prostate Cancer Cells

PONE-D-23-08305

Dear Dr. Kaczocha,

We’re pleased to inform you that your manuscript has been judged by both an independent reviewer and the Academic Editor..  Both believe that the manuscript is scientifically suitable for publication and will be formally accepted for publication once it meets all outstanding technical requirements.

Kind regards,

Salvatore V Pizzo

Academic Editor

PLOS ONE

Additional Editor Comments (optional):

After reviewing this manuscript, I concur with Reviewer #1, that this manuscript is acceptable for publication.

Reviewers' comments:

Reviewer's Responses to Questions

**Comments to the Author**

1. Is the manuscript technically sound, and do the data support the conclusions?

Reviewer #1: Yes

2. Has the statistical analysis been performed appropriately and rigorously? 

Reviewer #1: Yes

3. Have the authors made all data underlying the findings in their manuscript fully available?

Reviewer #1: Yes

4. Is the manuscript presented in an intelligible fashion and written in standard English?

Reviewer #1: Yes

5. Review Comments to the Author

Reviewer #1: In this work, the authors employed a taxane-resistant DU145-TXR cells and the parental Du145 cells to study whether and how suppressing FABP5 can increase the taxane treatment sensitivity of the high malignant Du145 prostate cancer cells. They demonstrated that FABP5 knockdown sensitizes the cells to docetaxel. In contrast, docetaxel potency was unaffected by FABP5 knockdown in taxane-sensitive DU145 cells. Taxane-resistance in DU145-TXR cells stems from upregulation of ABCB1. Expression analyses and functional assays confirmed that FABP5 knockdown in DU145-TXR cells markedly reduced ABCB1 expression and activity, respectively. Thus the authors concluded that this study demonstrated a potential new function for FABP5 in regulating taxane sensitivity and the expression of a major P-glycoprotein efflux pump in prostate cancer cells. The results are well presented and the experiments are well designed. The Du145 and Du145-TXR cells are excellent cell lines for conducting this investigation. The manuscript is now ready for publication.

6. PLOS authors have the option to publish the peer review history of their article (what does this mean?). If published, this will include your full peer review and any attached files.

Reviewer #1: No

---

## [Editor Report · Acceptance letter]

27 Sep 2023

PONE-D-23-08305 

Fatty Acid Binding Protein 5 Regulates Docetaxel Sensitivity in Taxane-Resistant Prostate Cancer Cells 

Dear Dr. Kaczocha:

I'm pleased to inform you that your manuscript has been deemed suitable for publication in PLOS ONE. Congratulations! Your manuscript is now with our production department. 

Kind regards, 

on behalf of

Dr. Salvatore V Pizzo 

Academic Editor

PLOS ONE